# Coarse-Grained Modeling of EUV Patterning Process Reflecting Photochemical Reactions and Chain Conformations

**DOI:** 10.3390/polym15091988

**Published:** 2023-04-22

**Authors:** Tae-Yi Kim, In-Hwa Kang, Juhae Park, Myungwoong Kim, Hye-Keun Oh, Su-Mi Hur

**Affiliations:** 1Department of Polymer Engineering, Graduate School, Chonnam National University, Gwangju 61186, Republic of Korea; taeyik97@gmail.com (T.-Y.K.); juhae0514@gmail.com (J.P.); 2Department of Applied Physics, Hanyang University, Ansan 15588, Republic of Korea; inhwa512@naver.com; 3Pritzker School of Molecular Engineering, The University of Chicago, Chicago, IL 60637, USA; 4Department of Chemistry and Chemical Engineering, Inha University, Incheon 22212, Republic of Korea; 5School of Polymer Science and Engineering, Chonnam National University, Gwangju 61186, Republic of Korea

**Keywords:** EUV photoresist, coarse-grained model, line edge roughness, stochastic effects, chain conformations

## Abstract

Enabling extreme ultraviolet lithography (EUVL) as a viable and efficient sub-10 nm patterning tool requires addressing the critical issue of reducing line edge roughness (LER). Stochastic effects from random and local variability in photon distribution and photochemical reactions have been considered the primary cause of LER. However, polymer chain conformation has recently attracted attention as an additional factor influencing LER, necessitating detailed computational studies with explicit chain representation and photon distribution to overcome the existing approach based on continuum models and random variables. We developed a coarse-grained molecular simulation model for an EUV patterning process to investigate the effect of chain conformation variation and stochastic effects via photon shot noise and acid diffusion on the roughness of the pattern. Our molecular simulation demonstrated that final LER is most sensitive to the variation in photon distributions, while material distributions and acid diffusion rate also impact LER; thus, the intrinsic limit of LER is expected even at extremely suppressed stochastic effects. Furthermore, we proposed and tested a novel approach to improve the roughness by controlling the initial polymer chain orientation.

## 1. Introduction

Photolithographic patterning technologies have undergone continuous development to reduce feature size. While multi-patterning with deep ultraviolet lithography (DUVL) has been introduced as a solution to the sub-10 nm node [1,2,3], the process can cause inefficiencies, including a high cost and low yield due to the long processing time. To avoid these problems, extreme ultraviolet lithography (EUVL) has been anticipated as an alternative patterning system capable of producing even finer features than DUVL with a single patterning process cycle using a 13.5 nm wavelength light source. Efforts for the cost-efficient high-volume manufacturing (HVM) of EUVL have focused on critical challenges of EUV optical source power, improvements in defects in both the mask, and resists and the trade-off relationship between resolution, line edge roughness (LER), and sensitivity (RLS trade-off) [4,5,6,7,8,9,10].

Chemically amplified resist (CAR) is a conventional EUV photoresist material that consists of a polymer matrix with protected groups, photoacid generators (PAGs), and quenchers. When exposed to EUV light, PAGs react with secondary electrons emitted by photons and release acids. The generated acids diffuse and change the solubility of functional groups during the post-exposure bake (PEB) step, transforming hydrophobic protected groups to hydrophilic deprotected groups. In positive-tone resists, the hydrophilic chains on the exposed area are dissolved by developers during development, leaving unsharp residual resists in the unexposed region that presents LER. On the other hand, in negative-tone resists, crosslinked polymer chains formed by the diffused acids remain after development. As the acid-catalyzed chemical reaction induced in the exposed resist area is a key factor in EUV CAR patterning, stochastic factors such as photon shot noise and secondary electron blur during exposure, and acid diffusion during the PEB process are known to significantly affect the roughness of the residual resist [11,12,13,14,15,16,17,18,19,20,21]. Efforts have been made to clarify the effect of acid generation and diffusion on the LER using a continuum model approach [16,17,22,23] and to capture the mobility of the catalyst and the deprotection kinetics through an atomistic molecular dynamics simulation [24,25].

However, as the feature sizes in the patterning process approach the molecular scale, variations in material distribution and polymer chain conformations during the patterning processes at the interface between exposed and unexposed areas become critical factors in pattern quality. In recent years, molecular structures and resist properties, including the molecular weight, thickness of the resist and glass transition temperature (*T_g_*) of polymer, have been also investigated as control parameters to improve LER [12,13,18,26,27,28]. Our previous work, which used a coarse-grained model, also demonstrated that molecular weight and *T_g_* are strongly correlated to LER formation, even when the stochastic effects of exposure and acid diffusion are excluded [27]. Koyama et al. investigated the combined effect of acid diffusion and degree of polymerization on the final pattern of both positive- and negative-type resists using molecular-scaled simulations, assuming an ideal aerial image in which exposed photons and emitted secondary electrons are only positioned in the exposed region [13]. They found that at a relatively low dose, the nonuniformity of acid distribution and deprotection reaction at the edge of the exposed region contribute most significantly to LER.

In this study, we considered an enhanced molecular simulation that took into account both the molecular and stochastic effects of a positive-tone photoresist in the EUV patterning process, aiming for a comprehensive understanding of resist systems affected by a combination of realistic stochastic factors and polymer conformation. We investigated the variation in final edge roughness when applying different photon distributions generated by the KLA PROLITH lithography simulator [29,30,31], various initial random chain conformations, acid diffusion lengths, and exposure doses. Additionally, we described the influence of initial chain alignment parallel to the interface on LER values as a novel way to possibly address the RLS trade-off.

## 2. Materials and Methods

We adapted diffusion Monte Carlo (MC) simulations with a theoretically informed coarse-grained (TICG) model and modified it to simulate the EUV lithography patterning process. The TICG model has been developed and validated to study the thermodynamic and kinetic properties of complex polymeric systems [32,33], particularly those involving microphase separation in block copolymers [34,35]. Building upon our previous work [27], we extended our model to account for the effects of polymer chain conformation as well as stochastic terms such as photon shot noise and acid diffusion. Figure 1 illustrates the overall patterning process of EUV lithography and snapshots of the simulation results of the developed model for each step: after post-applied bake (PAB); exposure to EUV light and acid generation; the acid-catalyzed deprotection reaction during post-exposure bake (PEB); after PEB; the before and after of the development; and after the drying process. Simulations were conducted within a 32 nm × 32 nm × 30 nm photoresist, considering 16 nm half-pitch patterning and its film thickness. The lateral direction was treated with periodic boundary conditions, while an impenetrable hard wall was used in the *z* direction. The initial photoresist was composed of polymer chains, PAG, and quenchers with ratios of 89.6%, 10%, and 0.4%, respectively. The polymer chains consisted of *N* coarse-grained beads of two different types: one representing repeated units containing protected functional groups and the other representing the backbone only. Protected beads were randomly distributed along the individual chain with a given ratio of functional groups, *f_p_* = 0.5. The PAG and quencher were also represented with beads having the same size as coarse-grained polymer beads. The bonded interaction with harmonic springs between adjacent beads in chains is defined as
$$\frac{H_{b}}{k_{B}T}=\frac{3}{2}\frac{N-1}{R_{e}{}^{2}}\overset{n}{\underset{k=1}{\sum }}\overset{N-1}{\underset{i=1}{\sum }}b_{k}{}^{2}\left(i\right)$$
where *k_B_* is the Boltzmann constant, *b_k_* is a vector connecting *i^th^* and (*i* + 1)*^th^* beads in a chain, and *R_e_* is the end-to-end distance of an ideal chain. Beads along each chain were initialized randomly on a sphere of radius of the average bond length, *b*_0_, which is fixed by the reference number of bead, *N* = 64. The non-bonded interaction as a function of local densities is described as

$$\frac{H_{nb}}{k_{B}T}=\int_{V}^{}dr\frac{\sqrt{\bar{N}}}{R_{e}{}^{3}}\left\{\chi_{AB}N\phi_{A}\phi_{B}+\chi_{BC}N\phi_{B}\phi_{C}+\chi_{AC}N\phi_{A}\phi_{C}+\frac{\kappa N}{2}{\left(1-\phi_{A}-\phi_{B}-\phi_{C}\right)}^{2}\right\}$$
where *ϕ_K_* is the normalized number density of type *K* particles in a cubic grid cell, *(*Δ*L)*^3^, $\sqrt{\bar{N}}$ is the average chain density that defines the number of materials with their ratio, *χ_αβ_N* is the segregation strength between type *α* and *β*, and *κN* is the inverse compressibility of the system. Δ*L*, $\sqrt{\bar{N}}$, and *κN* are fixed at 0.14 *R_e_*, 96, and 100, respectively. A, B, and C represent hydrophobic protected sites, the backbones of the copolymer, and PAGs/quenchers, respectively. In this work, *χN* values among A, B, and C were set to 0 as we assumed that those interactions were much weaker than the one between protected and deprotected sites. So, the non-bonded interaction was simplified as
$$\frac{H_{nb}}{k_{B}T}=\int_{V}^{}dr\frac{\sqrt{\bar{N}}}{R_{e}{}^{3}}\left\{\frac{\kappa N}{2}{\left(1-\phi_{A}-\phi_{B}-\phi_{C}\right)}^{2}\right\}$$

The initial random material arrangements were relaxed into the thermodynamically stable states through MC simulation with the bead displacement, mimicking the post-applied bake (PAB) after the spin coating. Figure 1a shows the side view of the equilibrated resist where red, yellow, purple, and dark-green beads indicate protected sites, the backbones of the copolymers, PAGs, and quenchers, respectively, after the PAB step. In the subsequent exposure step, the resist of relaxed chains, PAGs, and quenchers was subjected to photon distributions generated by PROLITH, a widely used commercial simulator modeling the optical and chemical aspects of photolithography [22,36,37,38,39]. The final pattern quality is attributed to complex photochemical reactions and various stochastic effects occurring during the processes. Any local variability that deviates from the ideal is referred to as a ‘’stochastic effect”, with the random and sparse photon distribution being known as the most influential factor. Using PROLITH, we generated three different density distributions of photons absorbed by the resist per cubic nanometer (nm^3^) along the metrology plane in the latent image before PEB, under an exposure condition of numerical aperture (NA) of 0.33, a dose of 34 mJ/cm^2^, an ASML dipole source, and the optimized illumination conditions. As the output of PROLITH is in the form of a continuous photon density field, we converted it into discrete photon counts in each grid cell by multiplying the photon density by the grid volume. The dimensions of the grid cell in the *x*, *y*, and *z* directions were 0.32 nm, 0.32 nm, and 0.3 nm, respectively. Figure 2 shows the generated three-dimensional stochastic photon distributions containing 1222, 1161, and 1204 photons, respectively. Acids that catalyze the deprotection reaction were generated via photochemical reactions with the quantum yield (*φ*) defined as the ratio of the generated acids per photon set to be 2. Among the PAG molecules located within 1 nm from a photon, we randomly selected PAG molecules numbering *φ* = 2 to react with the photon and be replaced with acid particles, as shown in Figure 3a, b. The material distribution in the resist containing generated acids (light green) by the first photon distribution is shown in Figure 1b.

After the exposure process, the resist underwent a post-exposure bake (PEB). During the PEB step, the generated acids are displaced by the diffusion length $\sqrt{6Ddt}$, which is governed by the diffusion constant (*D* = 0.2 nm^2^/s) and time interval (*dt* = 0.02) per MC cycle (Figure 3c). The reported acid diffusion coefficient varies widely depending on factors such as the resist constituent material and PEB temperature. However, in this study, we adopted the acid diffusion coefficient that had been previously found to yield the lowest LER in PROLITH simulations [22]. The diffused acids reacted with the nearest protected site (deprotection) or quencher (neutralization) within 0.2 nm in a single iteration step. The reaction parameters, which included the thermalization length, the deprotection/neutralization reaction radius, and diffusion constant, were chosen by referring to previous simulation studies of EUV lithography [22,40,41]. Polymer chains were set to be immobile because we assumed that the PEB temperature was lower than the glass transition temperature (*T_g_*) of the resist. A low number of 3000 simulation iterations was used to depict a standard PEB time of 60 s. Figure 3d,e show that the protected functional group (red) reacting with the acid changed to a deprotected site (blue), whereas the reacted quencher, which is represented by white particles, neutralized the acid. Figure 1c shows a snapshot of the resist after the completion of the PEB process.

After the deprotection/neutralization reaction, the developing solvents were placed on the resist with a 20% volume fraction of the resist for the development process (Figure 1d). The non-bonded interaction was expanded, considering the addition of the solvent particles, to represent the dissolution process:$$\frac{H_{nb}}{k_{B}T}=\int_{V}^{}dr\frac{\sqrt{\bar{N}}}{R_{e}{}^{3}}\left\{\chi_{AD}N\phi_{A}\phi_{D}+\chi_{AE}N\phi_{A}\phi_{E}+\chi_{DE}N\phi_{D}\phi_{E}+\frac{\kappa N}{2}{\left(1-\phi_{A}-\phi_{B}-\phi_{C}-\phi_{D}-\phi_{E}\right)}^{2}\right\}$$
where *A*, *B*, *C*, *D*, and *E* represent hydrophobic protected sites, the backbones of the copolymer, residual particles (i.e., acids, PAGs, and quenchers), hydrophilic deprotected sites, and the developer, respectively. *κN* = 100 was used in our simulation. We assumed that there were no favorable interactions between the backbone and functional groups (*χ_AB_N* = *χ_BD_N* = 0) between polymers and the PAG/quencher (*χ_AC_N* = *χ_BC_N* = *χ_CD_N* = 0), between the backbone and the developing solvent (*χ_BE_N* = 0), or between the developing solvent and the PAG/quencher (*χ_CE_N* = 0), while the protected functional groups and the deprotected sites had a repulsive interaction (*χ_AD_N* = 50). *χ_AE_N* and *χ_DE_N* were set to 300 and −200, respectively, to express the strong repulsive and attractive interactions between functional groups and developers. Solvated chains were replaced with developers when the average solvent density and minimum local solvent density were above 0.4 *beads/R_e_*^3^ and 0.2 *beads/R_e_*^3^. The PAGs and quenchers remaining after the PEB step were also replaced with the developers when the average solvent density around them was above 0.4 *beads/R_e_^3^*. The probability of accepting the MC move was determined by the difference in total energy between the initial and trial configurations and the weight factor as a function of solvent composition, as follows [27,42]:$$P_{acc}=\frac{1}{{\left(1+\phi_{O}{}^{8}\right)}^{4}{\left(1+\phi_{N}{}^{8}\right)}^{4}}\min\left[1,exp\left[-\frac{∆H}{k_{B}T}\right]\right]$$
where $\phi_{O}$ and $\phi_{N}$ are the compositions of the polymer and PAG/quencher at the original and trial configurations. The weight function suppressed the acceptance probability and decreased chain mobility at a high polymer concentration.

The dissolution of the resist was ceased when the critical dimension (CD) reached the target value of 16 nm (Figure 1e). Then, the residual resist went through the drying process (Figure 1f); solvents were replaced with the air, which was represented by F particles, where the segregation strength between polymers and air (*χ_AF_N = χ_BF_N = χ_CF_N*) was set to 400. Drying resulted in a collapse of the remaining chains. The *x* coordinates of the interface at the top view of the resist were tracked, and the line edge roughness (LER) was predicted by estimating 3*σ* deviation.

## 3. Results

### 3.1. The Effect of Photon and Material Distributions on LER

The LER values of the residual resist consisting of coarse-grained polymer chains of *N* = 64 were measured, and the results summarized in Figure 4a reflect these values when the photon distributions in Figure 2 were applied. To investigate the stochastic effect induced by various distributions of material components (polymer chains, PAG, and quencher molecules) on LER as well as the photon shot noise, the patterning procedures and calculation of 3*σ* were repeated for three initial random material conformations in the resist. In Figure 4a, the results of three different initial material configurations are marked in black, blue, and red, and three different photon distributions are indicated with squares, circles, and triangles; i.e., each color corresponds to a specific initial material configuration and each shape corresponds to a specific photon distribution. Appendix A compares the selected chain conformations at the interface and inside the unexposed area after the development process; after the developing solvent had been dried, collapsed chains were observed at the interface and relaxed chains were seen inside the resist. Figure 4 shows that even under an identical photon distribution, interfacial roughness varied considerably due to the variability in the initial random chain conformations. The variation in the material distribution caused LER to range from 3.2 nm to 4.1 nm, from 3.2 nm to 4.0 nm, and from 5.1 to 6.4 nm, for the first, second, and third photon distributions. The third photon distribution noticeably resulted in a larger LER regardless of the material distribution, implying that the stochastic effects from the photon distribution are a more dominant factor in determining LER than those from the randomness in the chain conformation. Additionally, we explored the effect of chain length on the roughness (Appendix A) and found that, contrary to our previous work, which showed that resists with a smaller molecular weight and high *T_g_* have a lower LER due to the smaller size of residual chains, causing them to collapse after drying [27], significant differences between the resists of different molecular weights were not found when stochastic effects were included in the simulations. We anticipate that chain length could become a more significant factor in reducing LER when the stochastic effect is suppressed, e.g., as would occur when a higher dose is used.

To quantify the impact of photons unexpectedly injected into an unexposed region on overall image contrast, we calculated the second moment (*s*^2^) of photon distribution in the unexposed region, located 4 nm from the interface. The second moment *s*^2^ was obtained by $\frac{1}{n}\Sigma X_{i}{}^{2}$, where *X_i_* is the number of photons positioned in the *i^th^* cell and n is the number of cubic cells with a side length of 2 nm. This cubic cell size was determined to be equivalent to the range of photoacid generation by a single photon. The third photon distribution resulted in the largest numbers of photon-occupied cells in the unexposed region with *s*^2^ = 0.340, while the first and second photon distributions showed an *s*^2^ of 0.281 and 0.271, respectively. Our findings suggest that the interfacial roughness in the system exposed to the third photon distribution can be attributed to its poor photon uniformity in the unexposed region. We also analyzed acid diffusion and the deprotection reaction during PEB steps for different photon distributions; however, we did not find significant qualitative differences in the acids distributions, as shown in Appendix A, which depicts the plot of the average acid variation (a–c) and deprotected sites (d–f) in the *x* direction during the PEB process (3000 MC iterations). However, undesired photons did induce a higher probability of a deprotection reaction occurring in the unexposed area, as demonstrated in Figure 4b. We marked the distributions of highly deprotected chains with a ratio of the number of deprotected sites, *f_depro_ = N_depro_/(N_pro_ + N_depro_)*, larger than 0.625 by placing spherical particles at the center of mass (*x_cm_*, *y_cm_*, *z_cm_*) of those chains in the first kind of resist (marked in black in Figure 4a). These particles were green at *f_depro_* = 0.625 and blue when the chain was fully deprotected. Highly deprotected chains were dissolved in developing solvents that were extremely attractive to deprotection groups during the development step. Therefore, the developers initially penetrated the middle part of exposed regions dominated by blue particles, and diffused laterally to dissolve the partially deprotected chains. The residual pattern after drying solvents, shown in red in Figure 4b, strongly correlates with the distribution of highly deprotected chain at the interface. Hence, the largest LER was observed from the third photon distribution, which presents a fuzzy distribution of highly deprotected chains in the unexposed region.

### 3.2. The Effect of Acid Diffusion, Quantum Yield, and Exposure Dose on LER

The sensitivity of LER to acid diffusion and exposure dose was tested by controlling the acid diffusivity, quantum yield (*φ*), and exposure dose with five sets of conditions, referred to as Cases I through V, which are listed in Table 1. Figure 5a shows the average deprotection fraction in the lateral direction after the PEB step for the three material configurations when the first photon distribution was applied. Case I is equivalent to the conditions in Figure 4.

In Figure 4 demonstrates that lowering the acid diffusivity in Case II retarded the deprotection reaction in the exposed region as well as the unexposed area, as noted by the red curve in Figure 5a, which failed to achieve the target CD. Increasing the quantum yield and slowing acid diffusion in Case III produced enough deprotected groups for patterning and resulted in a slight increase in the average deprotection fraction. In Case IV, the higher quantum yield without slowing acid diffusion elevated the production of deprotected groups in the exposed region, as indicated by the purple line in Figure 5a. Finally, in Case V, the higher exposure dose of 50 mJ/cm^2^ yielded the most enhanced contrast of deprotection fraction between the exposed and unexposed regions.

Figure 5b illustrates LER of the final patterns under different photochemical reaction conditions (Case I, III, IV, and V). In this figure, different colors and shapes of markers correspond to different material conformations and photon distributions. For Case I–IV, the resists exposed to the 34 mJ/cm^2^ EUV dose of the first and third photon distributions are marked in squares and triangles, respectively. For Case V, the roughness at 50 mJ/cm^2^ dose is represented by diamonds, circles, and stars, and the applied photon distributions are shown in Appendix A.

Case III, with a higher fraction of deprotected sites in the exposed region, resulted in a lower roughness than Case I. However, even in Case III, the resists with the third photon distribution had larger LER values compared to those exposed to the first photon distribution. Specifically, the bound of LER ranged from 2.64 nm to 3.56 nm for the first photon distribution (squares) and shifted up to 4.38 nm and 5.87 nm when the third photon distribution (triangles) was applied. In Case V, where a higher number deprotection sites was induced by the larger number of acids, LER further improved, ranging from 2.16 nm to 2.6 nm for the first photon distribution and from 3.66 nm to 4.44 nm for the third photon distribution. For Case V, in which the increased dose of 50 mJ/cm^2^ was used, the smallest roughness, ranging from 1.38 nm to 3.35 nm, was observed. The excellent LER reduction in Case V can be inferred from the high contrast in the deprotection fraction in the lateral direction. Furthermore, the 3D distributions of highly deprotected chains (*f_depro_* ≥ 0.625) after the PEB step and the final resist density in Figure 5c demonstrate more clearly that LER is strongly correlated to the distribution of deprotected chains and the degree of deprotection.

In Case IV and V, the exposed regions were occupied by a high concentration of fully deprotected chains (blue particles), allowing for the shortened penetration time of the developer, which interacted via a strong attraction to the deprotected sites with the resist of the unexposed area. Appendix A shows the variation in the dissolved polymer density in the *x* direction during the development process. However, despite the high concentration of fully deprotected chains and resulting fast development process in the center of the exposed area for both Cases IV and V, the distribution of fully deprotected chains in Case IV with a lower dose was much coarser at the interface, implying that the consequence of photon shot noise persists even under enhanced acid generation or diffusion conditions. It is notable that there was no clear distinction between the LER values of resists exposed to the three different photon distributions of 50 mJ/cm^2^, thus showing that a high dose reduces the effect of photon shot noise on LER. On the other hand, the longer development times required in Case I and III increased the chance of uneven chain dissolution and diffusion of developing solvents. Therefore, the photon distributions in the unexposed areas significantly affected the LER of the final pattern, resulting in higher interfacial roughness.

### 3.3. The Effect of Initial Chain Alignment on LER

Although high-dose EUV exposure can improve the interfacial roughness, it is not a cost-effective patterning method, and we still hope to develop an approach based on a low-dose system. As we have shown that the initial chain conformations in resists can significantly alter LER even when the same photon distribution is used, we further investigated the effect of initial chain orientations on final pattern roughness. We hypothesized that the influence of dissolved chains at the interface on the solvation of unexposed areas would be decreased when the chain conformation is parallel to the interface compared to when randomly or perpendicularly aligned chains are used.

To control the alignment of chains, we generated initial chain configurations by constraining the bond angle and x coordinate of beads. Starting from the first bead randomly placed on the chain, (*x*_0_*, y*_0_*, z*_0_), the next bead was placed at (*x*_0_*, y*_0_
*+ b*_0_*cosθ, z*_0_
*+ b*_0_*sinθ*), where *b_0_* is the average bond length of an ideal chain and *θ* is the bond angle. The bond angles *θ* are limited to the three different ranges of {[0, 2/3π) or (4/3π, 2π]}, {[0, 1/2π) or (3/2π, 2π]}, and {[0, 1/6π) or (11/6π, 2π]}. Chains with a narrower range of *θ* were more extended parallel to the interface. Subsequently, we applied short-chain relaxation steps that mimic the PAB process to release an artificially imposed chain stress during initialization. Even after the relaxation steps, alignment parallel to the interface was maintained, as shown in Appendix A. After development, the chains inside the resists remained stretched, while the chains at the interfaces in the residual patterns were in collapsed conformations due to strong repulsion between the air and polymers (Appendix A). The extent of chain stretching after the PAB process was quantified as the normalized radius of gyration, *R_g_/R_g,_*_0_, where *R_g,_*_0_ is the radius of gyration of an isotropically distributed chain. We averaged the measured *R_g_/R_g,_*_0_ over six cases of three different material distributions for both first and third photon distributions in Figure 2. The *<R_g_>/R_g,_*_0_ values for three different levels of chain alignments were 1.47, 1.78, and 2.0. Compared to the one-dimensional size of the isotropically relaxed chain *<R_g,_*_0*,x(y,z)*_> = 0.47 *R_e_*, the average *<R_g,y_*> increased to 1.01, 1.29, and 1.50 *R_e_*, and *<R_g,x_*> was reduced to 0.36 *R_e_* as the extent of parallel orientation was enlarged, indicating that the parallel configurations were retained after chain relaxation. The degree of chain alignment in the parallel direction was also quantified by measuring the second-rank order parameter, *P*_2_, after the PAB process [43]. *P*_2_ was obtained through:$$P_{2}=0.5\times \frac{1}{n}\times \frac{3}{N-1}\overset{n}{\underset{k=1}{\sum }}\left(\overset{N-1}{\underset{i=1}{\sum }}\cos^{2}\theta_{k,i}-1\right)$$
where *θ_k,i_* is the angle between the direction of the bond connecting the *i^th^* and (*i* + 1)*^th^* beads in the *k^th^* chain and the *yz* plane, *N* is the number of polymer beads per chain, and *n* is the total number of polymer chains. The *P*_2_ values after PAB were measured as 0.56, 0.58, 0.6, and 0.64, as *<R_g_>/R_g,_*_0_ was increased.

Figure 6a compares the final roughness of EUV patterning when the level of chain alignment was controlled, ranging from random (*<R_g_>/R_g,_*_0_ = 1) to highly stretched parallel to the interface (*<R_g_>/R_g,_*_0_ = 2). As the alignment level was increased to a higher *R_g_/R_g,_*_0_, the lowest LER values were reduced from 3.23 to 3.01, 2.49, and 2.40 nm for the first photon distribution and from 5.06 to 4.80, 4.65, and 3.43 nm for the third photon distribution. Improvement in the LER with stretching of the chain parallel to the interface is also demonstrated in Figure 6b, which shows the distribution of highly deprotected chains (colored center of mass of the chains with *f_depro_* ≥ 0.625) after the PEB step along with the residual pattern after the development process. Although the numbers of totally deprotected chains marked as blue particles were not large, an enhanced contrast in the distribution of highly deprotected chains over the resist was observed as the level of chain alignment increased. This enhanced contrast prevented the diffusion of the developer across the interfaces, which can cause the dissolution of chains in the unexposed area. As a result, a residual pattern with a sharper interface could be achieved. We tested the cases of the lowest LER achieved in Figure 6, that were exposed to the 1st photon distribution and had with a chain alignment of *<R_g_>/R_g,_*_0_ = 2.0, for the possibility of additional LER reduction. When we assumed chains during the PAB process were immobile, we achieved a further reduction in LER to 1.48 nm. Appendix A illustrates a comparison of chain conformations after the PAB step and projected views of the final residual pattern between the resists of mobile and immobile chains during the relaxation process. 

## 4. Conclusions

By developing a coarse-grained EUV patterning simulation model that takes into account both chain conformation and stochastic effects, including photon shot noise and acid diffusion, we investigated the variation in LER values of 16 nm patterns under various conditions. We compared the influence of the initial random material configurations and photon distributions on the roughness, finding that the LER of resists exposed to an identical photon distribution differed by 0.8–1.36 nm due to the randomness in the distribution of photoacid generators and chain conformations. There was a more significant difference of 1.06–3.22 nm among identical resist conformations when the photon distribution was varied. This difference was caused by the photons being unevenly located in the interface and unexposed region, which resulted in the uneven distribution of highly deprotection chains over the resist and eventually the formation of rough patterns. We found that an increase in the ratio of deprotected sites in the exposed region can reduce the roughness through studies on the effect of acid diffusion, quantum yield, and exposure dose on the final resists. At a low exposure dose, slowing acid diffusivity from 0.2 nm^2^/s to 0.1 nm^2^/s resulted in an insufficient deprotection reaction during the PEB step and a failure of 16 nm patterning. However, when the number of acids was increased through enhanced quantum yield, the deprotection reaction in the exposed region was improved and the final roughness was reduced. Applying a higher-dose EUV light to resists resulted in lower LER values and prevented the photon shot noise effect from occurring. Although the average distribution of acids or deprotected functional sites crossing the interfaces and the full 3D photon distribution itself did not show clear evidence of LER variation, we found that the 3D distribution of highly deprotected chains strongly correlated with the LER of final patterns. Our simulation further proposed an approach to reduce the LER, the control of initial chain orientation, which has been shown to be plausible with applying the shear force [44]. Greater parallel alignment of chains to the interface and the reduced variation in chain conformation during patterning led to a lower roughness. In a future study, we will aim to explore resists with a chain scission reaction and direct self-assembly of block copolymers combined with EUV patterns as alternative ways to improve LER.

## Figures and Tables

**Figure 1 polymers-15-01988-f001:**
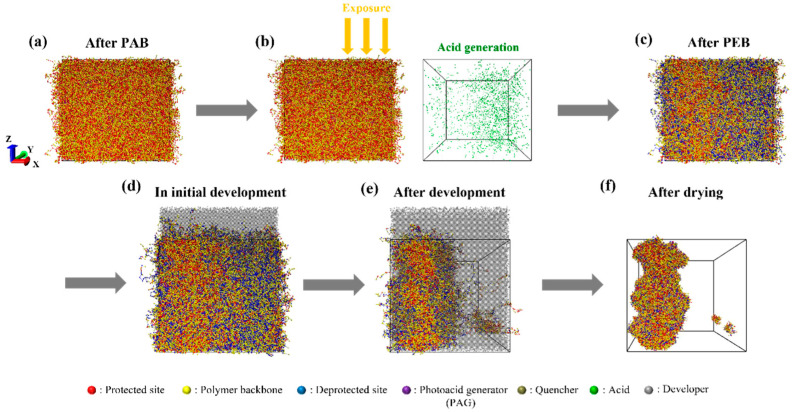
Side views of the simulation images of the resist of *N* = 64 and *f_p_* = 0.5 for each step over the EUV patterning. (**a**) After the chain relaxation via post-applied bake, (**b**) exposure and acid generation, (**c**) after post-exposure bake, (**d**) beginning of the development, (**e**) after the development, and (**f**) after the drying process. Red, yellow, blue, purple, dark-green, light-green, and gray particles represent protected functional groups, the backbone of polymers, deprotected sites, photoacid generators (PAGs), quenchers, acids, and developing solvents, respectively.

**Figure 2 polymers-15-01988-f002:**
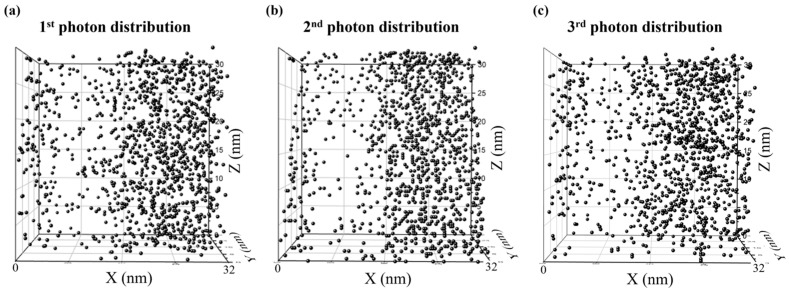
(**a**–**c**) Stochastic 3D photon distributions over 32 nm × 32 nm × 30 nm photoresist, obtained from PROLITH with 0.33 NA and 34 mJ/cm^2^.

**Figure 3 polymers-15-01988-f003:**
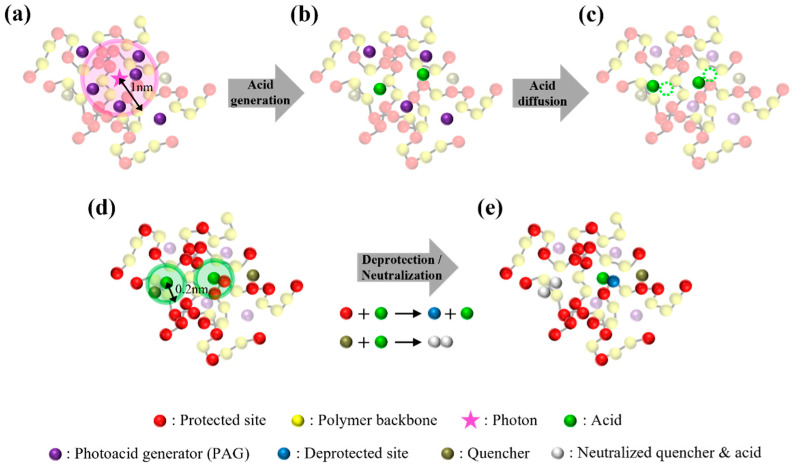
Schematic images of material changes after exposure to PEB in our model. (**a**,**b**) At quantum yield (*φ*) = 2, two photoacids (light green) were generated by replacing PAGs (purple) selected randomly within a 1 nm radius (pink range) of a photon (pink star). (**c**) The generated acids were displaced with the diffusion coefficient, *D* = 0.2 nm^2^/s. (**d**,**e**) The acids reacted with the closest particle within a 0.2 nm radius from the acid. The protected site (red) that reacted with the acid turned into a deprotected site (blue), while the acid that reacted with the quencher, which is represented by white particles, was neutralized.

**Figure 4 polymers-15-01988-f004:**
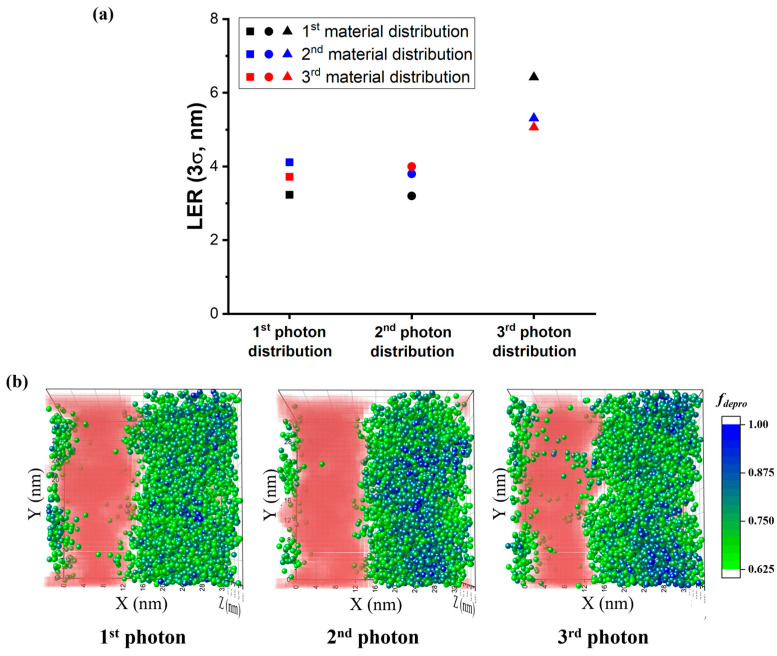
(**a**) The line edge roughness (LER) of the final patterns when photoresists of various material distributions were exposed to the three different photon distributions. The results of three different initial material configurations are marked in black, blue, and red. (**b**) The 3D distributions of highly deprotected chains (*f_depro_* ≥ 0.625) after the PEB step and the final resist’s density (red) after development. The 1st material configuration was exposed to the three different photon distributions. *f_depro_* is marked with differently colored spherical particles located in the center of mass of the corresponding chain (*x_cm_*, *y_cm_*, *z_cm_*).

**Figure 5 polymers-15-01988-f005:**
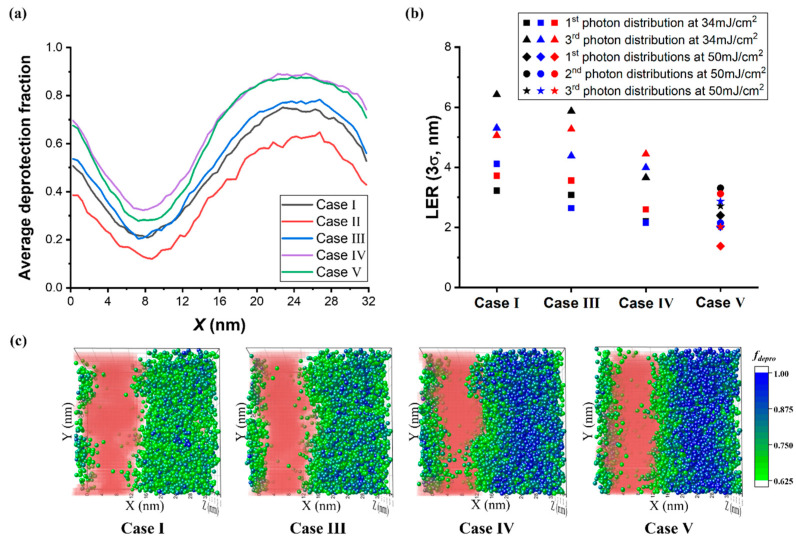
(**a**) Average deprotected site distribution in the *x* direction of the resist with different reaction conditions, Case I, II, III, IV, and V, which are represented by black, red, blue, and purple lines, respectively. The 1st photon distribution was applied to the resists at 34 mJ/cm^2^. (**b**) The roughness of the final patterns in Case I, III, IV, and V. Black, blue, and red markers correspond to different initial material conformations. The 1st and 3rd photon distributions of 34 mJ/cm^2^ were applied in Case I–IV, while three different photon distributions of 50 mJ/cm^2^ are applied in Case V. (**c**) The 3D distributions of chains with *f_depro_* ≥ 0.625 after the PEB step and the final patterns in Case I, III, IV, and V. The photoresists with the 1st material configuration were applied. The particle of *f_depro_* was located at the center of mass of the chain. The residual resist after drying the solvents is represented by red.

**Figure 6 polymers-15-01988-f006:**
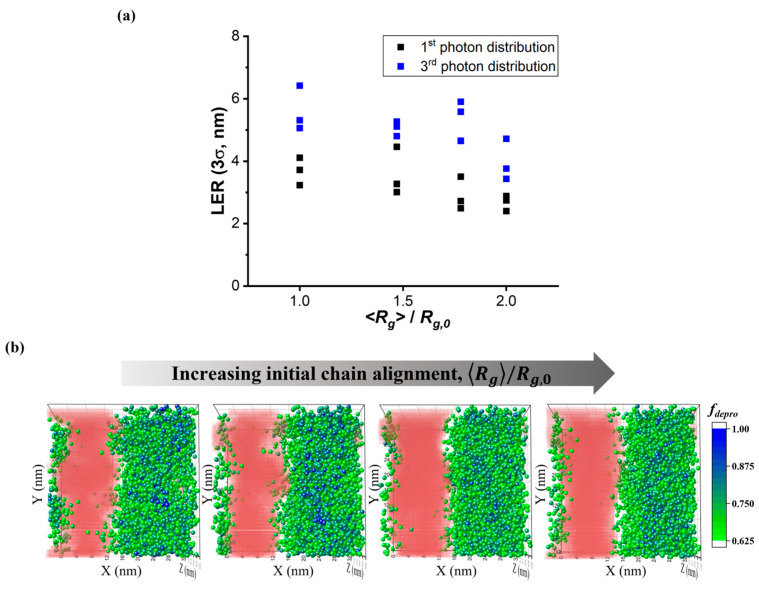
(**a**) The effect of the average extent of initial chain alignment parallel to the interface (<*R_g_>/R_g,_*_0_) on LER. *R_g_*_,0_ represents the radius of gyration of a random initial chain conformation after the PAB step. Photoresists with four different initial chain placements (<*R_g_>/R_g,_*_0_ = 1.00, 1.47, 1.78, 2.00) were exposed to the 1st (black) and 3rd (blue) types of photon distribution at a dose of 34 mJ/cm^2^. Initial bond angles for the cases of <*R_g_>/R_g,_*_0_ = 1.47, 1.78, 2.00 were set to be restricted over {[0, 2/3π) or (4/3π, 2π]}, {[0, 1/2π), or (3/2π, 2π]} and {[0, 1/6π) or (11/6π, 2π]}, respectively. (**b**) Variation in distribution of center of mass of chains with *f_depro_* ≥ 0.625 after the PEB step and final pattern led to an increase in <*R_g_>/R_g,_*_0_ from 1.00 to 2.00. The resists were exposed to the 1st photon distribution.

**Table 1 polymers-15-01988-t001:** Parameters for Case I–V: acid diffusivity, quantum yield, and exposure dose.

	Case I	Case II	Case III	Case IV	Case V
Acid diffusivity (nm^2^/s)	0.2	0.1	0.1	0.2	0.2
Quantum yield, *φ*	2	2	3	3	2
Exposure dose (mJ/cm^2^)	34	34	34	34	50

## Data Availability

The datasets generated during and/or analyzed during the current study are available from the corresponding author on reasonable request.

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
