# Peer review of "Coarse-Grained Modeling of EUV Patterning Process Reflecting Photochemical Reactions and Chain Conformations"

_polymers, 2023, doi:10.3390/polym15091988_

Round 1

Reviewer 1 Report

Coarse-grained Modeling of EUV Patterning Process Reflecting Photochemical Reactions and Chain Conformations

The manuscript has developed a coarse-grained model to investigate the effect of photon distribution, material distribution, acid diffusion, quantum yield and EUV dose on line edge roughness (LER) of chemically amplified resist patterning process. The authors also provided additional simulations to study the effect of polymeric chain alignment attempting to propose a novel way to control LER under the same exposure dose. This is a well-written manuscript describing a straightforward model to answer fundamental questions. The motivation and model details are clearly presented, and the results are well analyzed to support the conclusions. I recommend accepting this manuscript with a few minor points to be addressed.

Minor comments:

  • ‘pattering’, is this a typo?

  • Format of references: citations should be placed within the sentences.

  • Line 105 - 111: Reference for bonded and nonbonded potentials.

  • Line 154: references for the reaction parameters, e.g. reactive distance.

  • Line 229: ‘These results are … case.’ I’m not really following this sentence.

Materials and Setup:

  • Please elaborate on the variations of photon distributions. How are conditions controlled? Why is the difference significant?

  • Please elaborate on the variations of material distributions. If this is just repeated construction of model systems with different random seeds, you need a large sample size to quantify the impact of material randomness.

  • Experimental design: I think it is better to add a subsection here for experimental design. Currently the experimental design is embedded in Results making later discussion unclear.

Results:

  • Line 208 - 214: the sample size is not large enough to conclude this.

  • Line 218 - 219: please add explanation or hypothesis for this contradict.

  • Figure 5(b): the symbols of case V are not differentiable.

Author Response

We appreciate for reviewer’s questions and comments. Please see the attachment.

Reviewer 2 Report

See attachment.

Author Response

(The authors gave the same response as above.)
